# Superpixel-based Efficient Sampling for Learning Neural Fields from Large Input

## ABSTRACT

In recent years, novel view synthesis methods using neural implicit fields have gained popularity due to their exceptional rendering quality and rapid training speed. However, the computational cost of volumetric rendering has increased significantly with the advancement of camera technology and the consequent rise in average camera resolution. Despite extensive efforts to accelerate the training process, the training duration remains unacceptable for high-resolution inputs. Therefore, the development of efficient sampling methods is crucial for optimizing the learning process of neural fields from a large volume of inputs. In this paper, we introduce a novel method named Superpixel Efficient Sampling (SES), aimed at enhancing the learning efficiency of neural implicit fields. Our approach optimizes pixel-level ray sampling by segmenting the error map into multiple superpixels using the slic algorithm, and dynamically updating their errors during training to increase ray sampling in areas with higher rendering errors. Compared to other methods, our approach leverages the flexibility of superpixels, effectively reducing redundant sampling while considering local information. Our method not only accelerates the learning process but also improves the rendering quality obtained from a vast array of inputs. We conduct extensive experiments to evaluate the effectiveness of our method across several baselines and datasets. The code will be released.

## KEYWORDS

Neural Radiance Fields, Novel view synthesis, Large Input

## 1 INTRODUCTION

Novel view synthesis(NVS) has always been a hot research topic in computer vision, with wide-ranging applications in virtual reality, medical imaging, and augmented reality. In recent years, NVS methods that represent scenes using neural fields have gradually become prevailing due to their promising results [16–18, 21, 24, 41, 42, 46]. These methods represent the scenes with Deep Neural Networks (DNNs) and render the scenes through volume rendering. Compared with traditional Multi-View Stereo (MVS) methods [3, 5, 9, 11, 12, 40], they can easily handle textureless surfaces and interpolate unseen areas via regularization techniques [13]. However, the volume rendering of huge sampled points brings substantial

*ACM MM, 2024, Melbourne, Australia*

© 2024 Copyright held by the owner/author(s). Publication rights licensed to ACM.
ACM ISBN 978-x-xxxx-xxxx-x/YY/MM
https://doi.org/10.1145/nnnnnnn.nnnnnnn

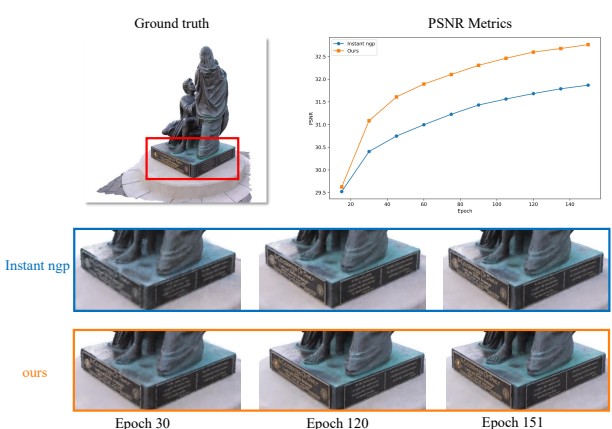

**Figure 1: Comparison between Instant NGP and Our Method: The top right corner displays PSNR curves for both methods on the test set, while the visualization below showcases their rendering results. Incorporating our method enhances the convergence speed and rendering quality of Instant NGP noticeably.**

computational overhead, which becomes even more severe with increasing image resolution.

For novel view synthesis, the resolution of input images directly determines the upper limit of reconstruction quality. With the advancement of camera technology, the resolution of captured images continues to increase. However, high-resolution images not only capture more details but also bring a substantial amount of redundant data. Therefore, traditional random pixel sampling methods [20, 34] become overwhelmed by the huge number of pixels in high-resolution inputs. [44] employs a patch sampling strategy combined with an encoder-decoder structure. They use the encoder to encode 3D geometric information and the decoder to render high-resolution outputs. [35] introduces color-guided ray sampling. They guide the sampling of rays by color and depth, allowing the rays to pass through regions with richer spatial information. However, their method does not simultaneously consider both color and rendering errors, and the redundant sampling of rays results in slower convergence on high-resolution images.

To address the challenge of large input, we introduce a Superpixel-based Efficient Sampling (SES) method for learning neural fields. Our core idea is to increase the number of light paths to acquire more information, especially in regions with significant rendering errors, thereby reducing redundancy in sampling. Simultaneously, we aim to optimize the sampling of light rays to consider local image information. We first employ the SLIC algorithm to segment the input image into different superpixels, where each superpixel represents distinct local information. Compared to other segmentation

methods like quadtree, superpixels can significantly reduce redundancy. In subsequent steps, we treat superpixels as the smallest processing unit. We set different sampling probabilities for different superpixels based on their rendering errors, continuously update them during training, and incorporate the rendering errors into the loss function. We apply our method to several popular NeRF algorithms and test it on multiple common datasets. Our image rendering quality reached the state of the art at different resolutions, and integrating our method did not incur additional computational overhead. In summary, our main contributions can be summarized as follows:

- We propose an efficient and effective sampling method for learning neural fields from large input. By segmenting the image into superpixels, we reduce light ray redundancy, improve sampling efficiency, and take into account local sampling information.
- We have devised a new loss function to expedite the convergence of the network in regions with significant rendering errors.
- We conduct comprehensive experiments based on popular neural field methods across several datasets, demonstrating the advantages of the proposed method.

## 2 RELATED WORK

### 2.1 Novel view Synthesis

In graphics research, the pursuit of Novel View Synthesis (NVS) hinges on adeptly handling intermediate 3D scene representations. Previous endeavors have focused on exploring suitable 3D representations: Mesh-based or pointcloud-based methods[2, 4, 6, 27, 39] often rely on depth information or structure from motion(SFM)[28] techniques for geometric shape restoration, presenting challenges in rendering realistic images. While approaches utilizing multi-plane images(MPIs)[7, 19, 32, 33, 38, 51] to express scenes as feature maps or stacked images can yield high-quality renderings, they constrain the variability of perspectives. Voxel-based representations[14, 17, 25, 29, 30, 36] offer swift image rendering but face limitations in resolution. By modeling scenes as neural implicit fields[20, 22, 31, 47], greater flexibility is attained alongside high-quality image generation: Differentiable Volume Rendering (DVR)[23] and Implicitly Differentiable Renderers (IDR)[48] have demonstrated promising results, albeit still requiring precise object mask inputs during training. The year 2020 saw the groundbreaking introduction of NeRF[20], revolutionizing NVS through volume rendering. NeRF employs neural networks to model spatial 3D coordinates and view directions as volume density and color, employing alpha blending for pixel color rendering, resulting in remarkable outcomes. NeRF's advent has significantly advanced the domains of new view synthesis, 3D reconstruction, point cloud rendering, and beyond.

### 2.2 Training acceleration

NeRF employs volume rendering to render pixel colors, and to bring sampling points closer to the surface, it introduces a two-stage sampling method along the rays. However, such an approach typically demands 1-2 days or even several days for training completion. To expedite NeRF training, ENeRF[10] utilizes the geometric features of the scene to guide surface sampling. Some methods leverage explicit modeling techniques to accelerate scene training: kilo NeRF[26] decomposes NeRF into multiple small MLPs for faster training, while DVGO[34], Plenoxels[8], and similar works employ learnable explicit meshes to expedite training. Instant ngp[21] utilizes multi-resolution hash encoding with a hash table and jump sampling to swiftly approach the scene surface during training. The aforementioned works focus on accelerating sampling strategies in space; Wang et al.[50] use a quadtree to guide sampling, reducing the number of sampled rays to accelerate training, while Sun et al[35]. efficiently sample rays by filtering them using images and depth information. These methods greatly accelerate NeRF training, although they may encounter limitations when high-resolution images are input.

### 2.3 High-resolution Synthesis

To handle high-resolution image inputs and generate high-resolution images, 4k-NeRF[44] encodes high-resolution images into feature maps. After training, a decoder is used to decode these feature maps into high-resolution images. This method reduces the computational burden during NeRF training. However, joint training of the encoder-decoder with NeRF still requires a considerable amount of time. UHD-NeRF[15] reconstructs higher-resolution images by using a combination of neural networks and point cloud representations. It intends to reconstruct the low-frequency components of the scene using MLP and the high-frequency components using point clouds. It is worth noting that both methods mentioned above use patch sampling to avoid excessive memory consumption. While patches can capture local information in the scene, their fixed shape may lead to fragmented image information.

## 3 METHOD

Our primary objective is to address the slow convergence and scattered sampling issues brought about by high-resolution inputs, ensuring superior rendering results within the same training timeframe. Firstly, in Section 3.1, we introduce the NeRF[20] and SLIC[1] segmentation algorithm, forming our work's foundation. Subsequently, in Section 3.2, we present our two-stage segmentation method and how we utilize superpixels to guide sampling. Finally, in Section 3.3, we demonstrate how our method integrates into existing baselines. Figure 2 illustrates our entire pipeline.

### 3.1 Preliminaries

**NeRF:** In NeRF[20], a 3D scene as a continuous function which takes as input 3D position $x = (x, y, z)$ and viewing direction $d = (\theta, \phi)$ and predicts the radiance color $c$ and volume density $\sigma$. The architecture of $f_\theta$ is chosen such that only the color $c$ depends on the viewing direction $d$. This allows the modeling of view-dependent effects like specularities and reflections while also encouraging a consistent geometry to be learned. In a deterministic pre-processing step $x$ and $d$ are transformed by a positional encoding $\gamma$ which promotes learning of high-frequency details. NeRF is typically parametrized by a multilayer perceptron (MLP) $f : (\gamma(x), d) = (c, \sigma)$. To render a pixel by a given camera pose, the expected color $\hat{C}(r)$ of a camera ray $r = o + td$ through the pixel

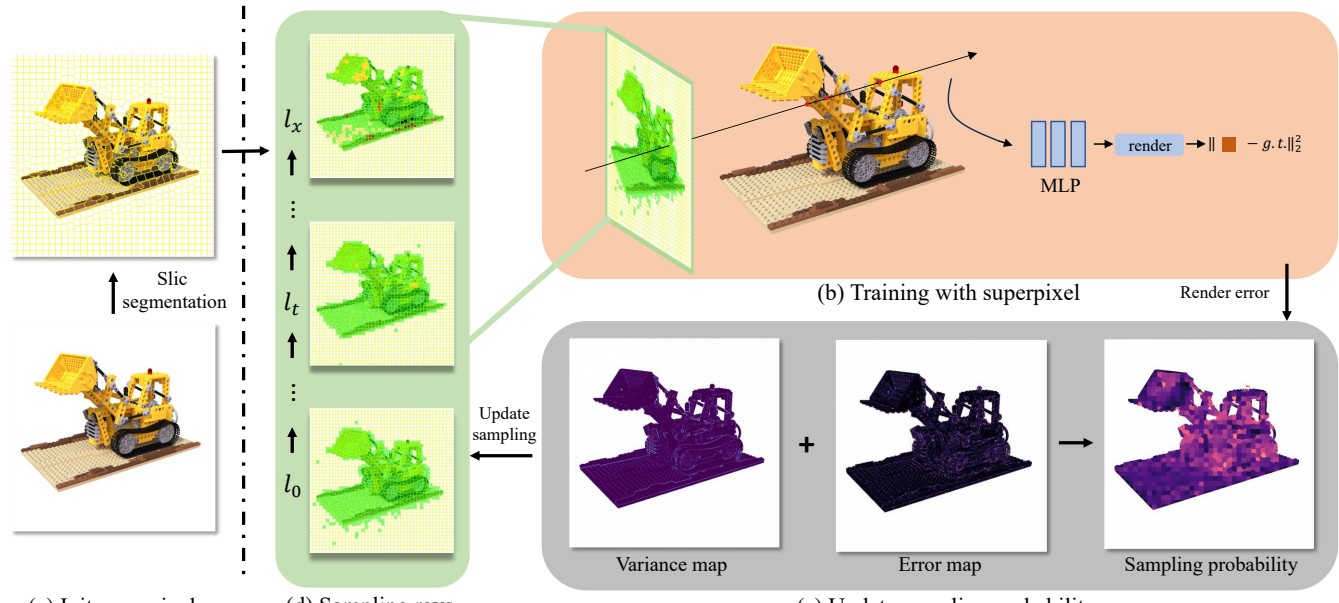

(a) Init superpixel          (d) Sampling rays                    (c) Update sampling probability

**Figure 2: Overview of our pipeline. (a) Firstly, we initialize the segmentation of input images using the SLIC algorithm. After a period of training, secondary segmentation is conducted at certain locations, yielding the final superpixels, and generating sampled rays. (b) The sampled rays are trained and the rendering error is recorded after each session for subsequent updates. (c) After ray sampling training, the rendering error for each superpixel is computed based on accumulated data. A sampling probability map is generated by combining pixel variances, and sampled rays are regenerated accordingly.**

that shots from the camera $o$ in direction $d$ can be calculated as

$$C(\mathbf{r}) = \int_{t_n}^{t_f} T(t)\sigma(\mathbf{r}(t))\mathbf{c}(\mathbf{r}(t), \mathrm{d})\mathrm{d}t \qquad (1)$$

where

$$T(t) = exp(-\int_{t_n}^{t_f} \sigma(r(t))dt) \qquad (2)$$

the accumulated transmittance indicates the probability that a ray travels from $t_n$ to $t$ without hitting any particle. NeRF is trained to minimize the mean-squared error(MSE) between the predicted renderings and the corresponding ground-truth color:

$$\mathcal{L}_{\mathrm{MSE}} = \sum_{\mathbf{p} \in \mathcal{P}} \|\hat{C}(\mathbf{r_p}) - C(\mathbf{r_p})\|_2^2 \qquad (3)$$

where $\mathcal{P}$ denotes all pixels of training set images, $\hat{C}(r_p)$ and $C(r_p)$ are the ground truth and output color of $p$.

**SLIC:** Simple Linear Iterative Clustering(SLIC)[1] is a fast and effective algorithm for image segmentation. As shown in algorithm1 combines superpixel methods with K-means clustering to cluster pixels in the color space, thus achieving image segmentation. SLIC first grids pixels into compact superpixels and then groups these superpixels into segments based on pixel similarity and distance measurements. By controlling superpixels' compacity and distance parameters, SLIC can significantly reduce computational complexity while preserving image details. Furthermore, SLIC excels in preserving boundary information of the image, allowing the segmentation results better to retain the sharpness and continuity of

---

**Algorithm 1** SLIC superpixel segmentation

1: /* Initialization */
2: Initialize cluster centers $C_k = [l_k, a_k, b_k, x_k, y_k]^T$ by sampling pixels at regular grid steps $S$.
3: Move cluster centers to the lowest gradient position in a 33 neighborhood.
4: Set label $l(i) = -1$ for each pixel $i$.
5: Set distance $d(i) = \infty$ for each pixel $i$.
6: **repeat**
7:     **for** each cluster center $C_k$ **do**
8:         **for** pixel $i$ in a $2S * 2S$ region around $C_k$ **do**
9:             Compute the distance $D$ between $C_k$ and $i$.
10:            **if** $D < d(i)$ **then**
11:                set $d(i) = D$
12:                set $l(i) = k$
            **endif**
        **endfor**
    **endfor**
13:    /* Update */
14:    Compute new cluster centers.
15:    Compute residual error $E$.
16: **until** $E \leq$ threshold

---

object boundaries. The distance between $C_k$ and i is calculated by

the following formula:

$$D = \sqrt{(L - L_i)^2 + (a - a_i)^2 + (b - b_i)^2 + \left(\frac{S - S_i}{S_N}\right)^2} \quad (4)$$

where $L, a, b$ are the values in the LAB color space of the pixel, $S$ is the position of the pixel, $L_i, a_i, b_i$ are the values in the LAB color space of the center of the superpixel block, $S_i$ is the position of the center of the superpixel block, $S_N$ is the normalization factor, equal to image size divided by the number of superpixel numbers.

## 3.2 Superpixel guide sampling

NeRF[20] perceives space by learning a color-constrained network to output color and density information of three-dimensional spatial points. Compared to simple regions in space, complex regions require more iterations for the network to better learn spatial information. Previous methods[35, 50] have attempted multiple sampling in regions with large rendering errors or drastic color changes while reducing sampling in areas with small rendering errors to achieve better results. Redundant computations resulting from insufficient partitioning by quadtrees and other segmentation methods increase continuously with input scale and are not feasible. Methods guiding computation through color gradients also struggle with handling large-scale image blocks. To address these issues, we use the SLIC[**?**] algorithm to segment images into different superpixels, which serve as the smallest processing units for guiding sampling. Due to the characteristics of superpixels, the problem of insufficient partitioning caused by structures like quadtrees is resolved, reducing computational overhead in single processing instances.

**Image to superpixel** During the training process, we will update the SLIC segmentation results twice, which enables the segmentation to focus on rendering errors while conforming to geometric constraints. At the beginning of training, we will employ the SLIC algorithm to partition the input RGB image into superpixels.

$$\ell^i = F(T_{CIE}(Image)), \ell^i \in L \quad (5)$$

where $\ell_i$ represents the $i$th superpixel in the image, $L$ denotes the collection of all superpixels segmented in the image, $F$ represents the SLIC algorithm1, with its process outlined in Algorithm 1, and $T$ signifies the transformation of the image from the RGB color space to the ICE color space.

Then, during the training process, we will update the superpixel segmentation results for the second time to make the segmentation more focused on areas with larger rendering errors. However, it is not necessary to update all superpixels to avoid disrupting the original segmentation structure. Refer to Figure 3, we will concentrate the re-segmentation effort on the areas with the largest rendering errors and their adjacent areas, using the masked slic algorithm to re-segment these regions. In the re-segmentation process, both RGB color and rendering error are considered simultaneously. First, the rendering error is mapped to the RGB color space, and then it is converted to the CIE color space. During the distance calculation with the SLIC algorithm, the distances in both the RGB and CIE color spaces are considered.

$$C_k = \beta_0 T_{CIE}(I) + (1 - \beta_0)T_{CIE}(R) \quad (6)$$

Where $I$ represents the portion of the input RGB image that needs to be re-segmented, $R$ represents the portion of the render error

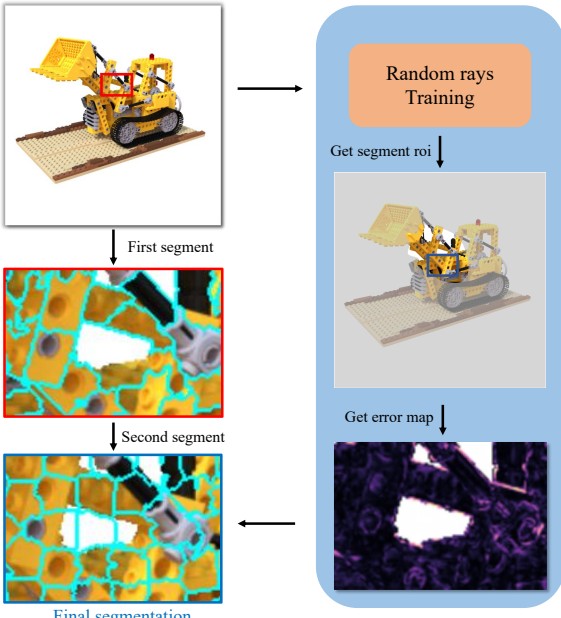

**Figure 3: Two rounds of segmentation in our approach: firstly, an initial segmentation is performed on the RGB image. Following a training period, areas necessitating re-segmentation are identified, and re-segmentation is carried out by considering both rendering errors and RGB colors.**

that needs to be recalculated. This portion is replicated three times to meet the dimensional requirements for the transformation to the ICE space. $\beta_0$ is a regularization term used to adjust the weights between the RGB image and the render error.

**Sample with superpixel** We employ superpixels to guide the pixel-level sampling during the training process. Initially, we compute the average sampling error within each superpixel. Each superpixel's error $\ell_{error}^i$ is computed as follows:

$$\ell_{error}^i = \frac{1}{N} \sum_{j \in N} r_j \quad (7)$$

where $N$ represents the number of pixels sampled in this segment, and $r_j$ denotes the rendering error of the current pixel.

We will calculate the sampling probability for each superpixel based on its rendering error and RGB color. Initially, we will exclude superpixels with negligible errors, corresponding to either blank areas in the scene or well-learned regions. Rendering error serves as the primary indicator for sampling probability calculation. However, due to significant variations in rendering error scales within the scene, sampling tends to concentrate on a few superpixels. To ensure comprehensive scene coverage, we employ the square root of the error as a representation of sampling probability. Despite the stochastic nature of sampling, sampling error may not fully capture sampling probability. Thus, we supplement this with RGB information. By computing the variance of colors within each superpixel, we collectively determine the sampling probability for each superpixel. Therefore, the sampling probability $g(\ell_{error}^i)$ for

each superpixel can be calculated as follows:

$$\ell^i_{error\_p} = \begin{cases} \dfrac{clamp(t, \sqrt{\max(\ell^i_{error})})}{\sqrt{\max(\ell^i_{error})}}, & if \quad \ell^i_{error} > t \\ 0, & if \quad \ell^i_{error} <= t \end{cases} \quad (8)$$

$$\ell^i_{std\_p} = \sqrt{\frac{1}{m} \sum_{x,y} [\mathbf{c}(x,y) - \bar{\mathbf{c}}]^2}, \quad x,y \in \ell^i. \quad (9)$$

$$\ell^i_p = \lambda_{error} \cdot \ell^i_{std\_p} + \lambda_{std} \cdot \ell^i_{error\_p} \quad (10)$$

where $t$ denotes the minimum error threshold we set. We normalize the sampling probability of the rendering error component to lie between 0 and 1, $m$ represents the number of pixels in the current superpixel, and $\bar{\mathbf{c}}$ denotes the average color within the superpixel. $\lambda_{error}$ and $\lambda_{std}$ are the coefficients for the sampling probabilities. We adjust these coefficients to balance the weights of the two probabilities.

## 3.3 Superpixel guide sampling

We integrate superpixel sampling into NeRF training and employ a series of strategies to adapt to high-resolution inputs, thereby enhancing the rendering quality of NeRF.

**Training Progress.** Our training process is illustrated in Figure 2. Initially, we employ a random sampling method for training while updating the errors of superpixels in each training iteration. After the secondary segmentation computation, we utilize superpixels to guide sampling and ray generation, concurrently accumulating rendering errors. Upon completion of training for all rays generated in a single sampling, we update the superpixel errors and regenerate rays.

**Loss Design.** We optimize the parameters of the MLP using pixel-level Mean Squared Error (MSE) loss $\mathcal{L}_{mse}$ and superpixel loss $\mathcal{L}_{sp}$. Our total loss is computed as follows:

$$\mathcal{L} = \lambda_{mse} \cdot \mathcal{L}_{mse} + \lambda_{sp} \cdot \mathcal{L}_{sp} \quad (11)$$

The calculation method of $\mathcal{L}_{sp}$ is as follows:

$$\mathcal{L}_{sp} = \ell^i_{error} + 0.001 \cdot \ell^i_{std\_p} \quad (12)$$

Due to the strong bias of $\mathcal{L}_{sp}$, $\lambda_{sp}$ is typically set to be three to four orders of magnitude smaller than the $\lambda_{mse}$ during the training process.

**Progressive Training.** In NeRF[20] and its related works, position encoding is utilized to enable MLPs to learn more high-frequency information. However, in practical tests, we found that while higher frequency position encoding captures more detailed information, it leads to slower convergence. In our work, we focus on enabling the training to rapidly learn the scene's low-frequency information in the initial, low-frequency stages. Therefore, we adopt a strategy similar to that used in NeuS2[42] and FreeNerf[45], gradually increasing the bandwidth of the position encoding during the training process. In NeRF, the representation of position encoding is as follows:

$$\gamma_L(\mathbf{x}) = \left[ \sin(\mathbf{x}), \cos(\mathbf{x}), ..., \sin(2^{L-1}\mathbf{x}), \cos(2^{L-1}\mathbf{x}) \right] \quad (13)$$

$$\gamma'_L(\mathbf{x}) = \gamma_L(\mathbf{x}) * M(i) \quad (14)$$

where, $L$ represents the maximum frequency in position encoding, and $\gamma_L(\mathbf{x})$ defines the specific formulation of position encoding.

The position encoding used in our work is $\gamma'_L(\mathbf{x})$ which is formed by multiplying $\gamma_L(\mathbf{x})$ with $M$ which is a mask with the same dimension as $\gamma'_L(\mathbf{x})$, designed to mask the high-frequency components, with its specific parameters controlled by the number of input iterations $i$. In practical training, we set the maximum frequency of the position encoding to a quarter of its original value, which is gradually increased as the training progresses and the resolution enhances.

## 4 EXPERIMENTS

### 4.1 Implementation Details

In comparison to structures like quad-trees, superpixels do not possess similar tree-like structures to minimize sampling redundancy. To fully leverage hardware performance, we provide our method's implementation on CUDA. In our implementation, for synthetic datasets, we set $\beta_0$ to 0.8 as per Equation 6, $\lambda_{mse}$ as 1, $\lambda_{sp}$ as 1e-6, whereas for real datasets, we choose $\beta_0$ as 0.65, $\lambda_{mse}$ as 1, $\lambda_{sp}$ as 1e-8. Our method can be seamlessly integrated into the popular NeRF framework. To demonstrate the efficiency of our approach, we selected Instant-NGP[21] and DVGO[34] as our baselines, both known for their fast training speeds within the NeRF domain. It is worth noting that, in our experiments, we utilized an open-source implementation of Instant NGP, namely the nerf-template, which is a clean version derived from the open-source project torch-ngp[37]. We employed this implementation in our experiments as a replacement for the original Instant NGP. We conducted experiments using a single NVIDIA RTX 4090 GPU.

### 4.2 Dataset

We tested our method on the currently popular NeRF dataset to demonstrate its superiority. We conducted experiments on three datasets: eight synthetic 360° scenes provided by NeRF, four scenes from Tank and Template, and eight real scenes from MipNeRF 360 dataset. We also provided ablation experiments to validate our method. We evaluate the quality of view synthesis for ground truth from the same pose using three metrics: PSNR, SSIM[43], and LPIPS[49].

### 4.3 Realistic Synthetic 360 evaluation

The Realistic Synthetic 360 evaluation dataset[20] is a commonly used benchmark in the NeRF field. Using Blender, it renders eight scenes, each with distinct characteristics, and generates 100 training images per scene, making it a popular choice for multi-view reconstruction benchmarks. While the native renderings are at a resolution of 800x800, based on the provided Blender files, we generated images at higher resolution(1600x1600, 3200x3200). These newly rendered high-resolution images maintain the original viewing angles but offer enhanced clarity. Due to memory constraints, we conducted experiments only at resolutions of 800 and 1600 on DVGO.

The quantitative results of our method are presented in Table 1. We compare our approach with the baseline at different scales, revealing superior performance across all resolutions, notably in Hotdog and Lego scenes. Visualizations of our results are illustrated in Figure 4, demonstrating enhanced scene details and reduced

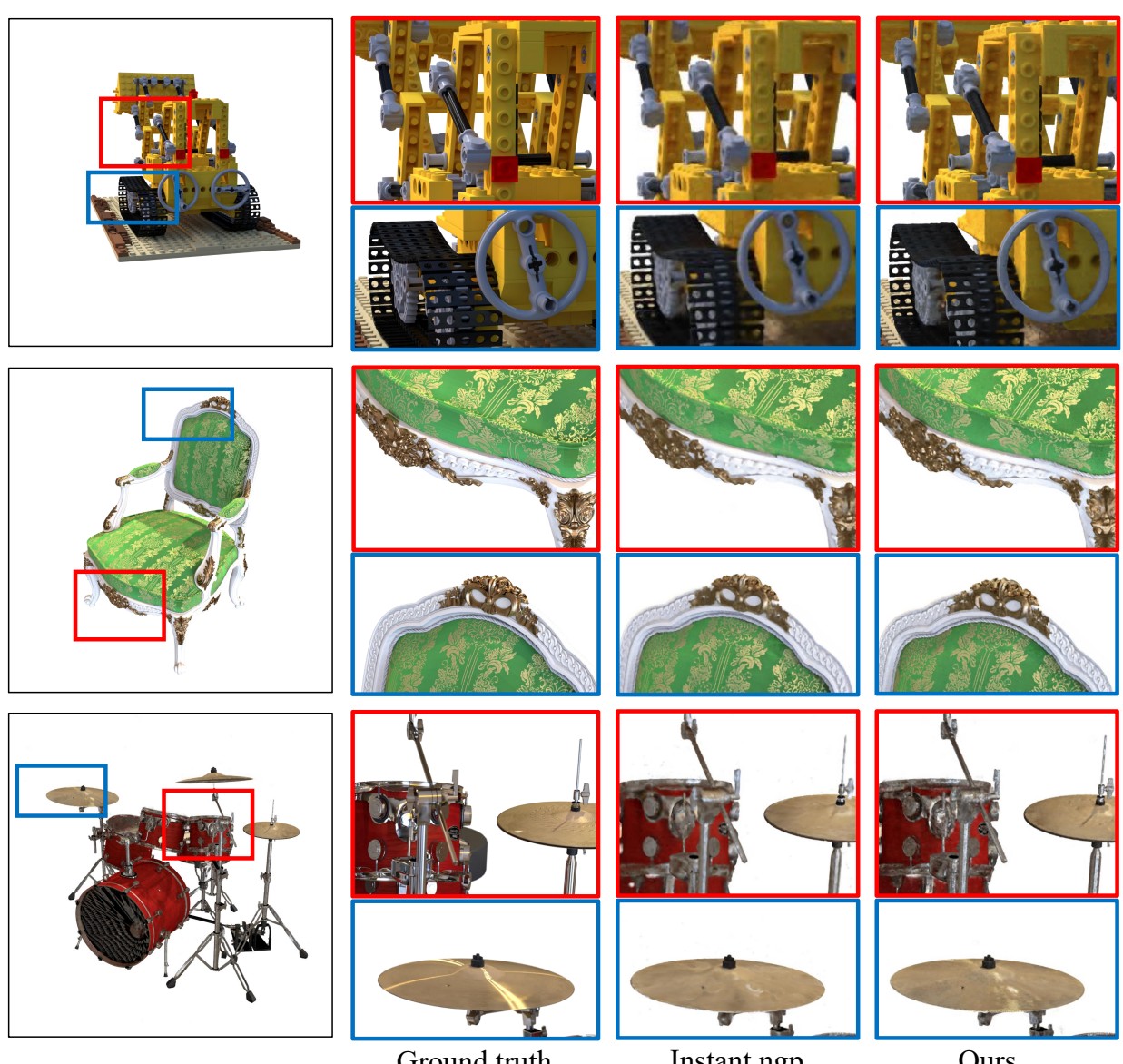

Ground truth     Instant ngp     Ours

**Figure 4: Results of NeRF Synthetic 3200x3200. The figure showcases the results of our method combined with Instant NGP on high-resolution inputs from the NeRF synthetic dataset. With the incorporation of our method, the rendered images exhibit increased detail and high-frequency information.**

artifacts, particularly evident in scenarios with large-scale inputs, upon integration of our method.

## 4.4 MipNeRF 360

MipNeRF 360 is a commonly used real-world unbounded scene dataset, comprising intricate scenes without boundaries. We conducted experiments on seven distinct scenes within this dataset. Each scene dataset includes original 4K images alongside their corresponding downsampled versions. We evaluated the 2x and 4x downsampled input images. Results demonstrate that our proposed

approach outperforms baseline models, particularly excelling in high-resolution scenes.

In Table 2, we present the quantitative results of our experiments, demonstrating that our method adapts well to both resolutions and consistently improves performance over the selected baselines. Our visual results are showcased in Figure 5, where it can be observed that our method reconstructs more details compared to the baselines, particularly evident with high-resolution inputs. We also monitored the impact of incorporating our method on the training time of the baseline. The addition of our method results in only a

(a) NeRF Synthetic.

| Method | Blender 800x800 | | | Blender 1600x1600 | | | Blender 3200x3200 | | |
|---|---|---|---|---|---|---|---|---|---|
| | PSNR↑ | SSIM↑ | LPIPS↓ | PSNR↑ | SSIM↑ | LPIPS↓ | PSNR↑ | SSIM↑ | LPIPS↓ |
| DVGO | 31.77 | **0.955** | **0.055** | 30.30 | 0.942 | 0.087 | - | - | - |
| DVGO(ours) | **31.88** | **0.955** | 0.057 | 30.57 | **0.951** | **0.086** | - | - | - |
| DVGO(wo mask) | 31.15 | 0.951 | 0.062 | 30.06 | 0.940 | 0.090 | - | - | - |
| instant ngp | 30.68 | 0.945 | 0.078 | 29.79 | 0.935 | 0.108 | 28.72 | 0.942 | 0.084 |
| instant ngp(ours) | 31.11 | 0.948 | 0.078 | **30.99** | 0.942 | 0.088 | **29.28** | **0.944** | **0.080** |

**Table 1: These are the quantitative results for the NeRF synthetic dataset. As demonstrated, our method shows performance improvements in the majority of the scenes.**

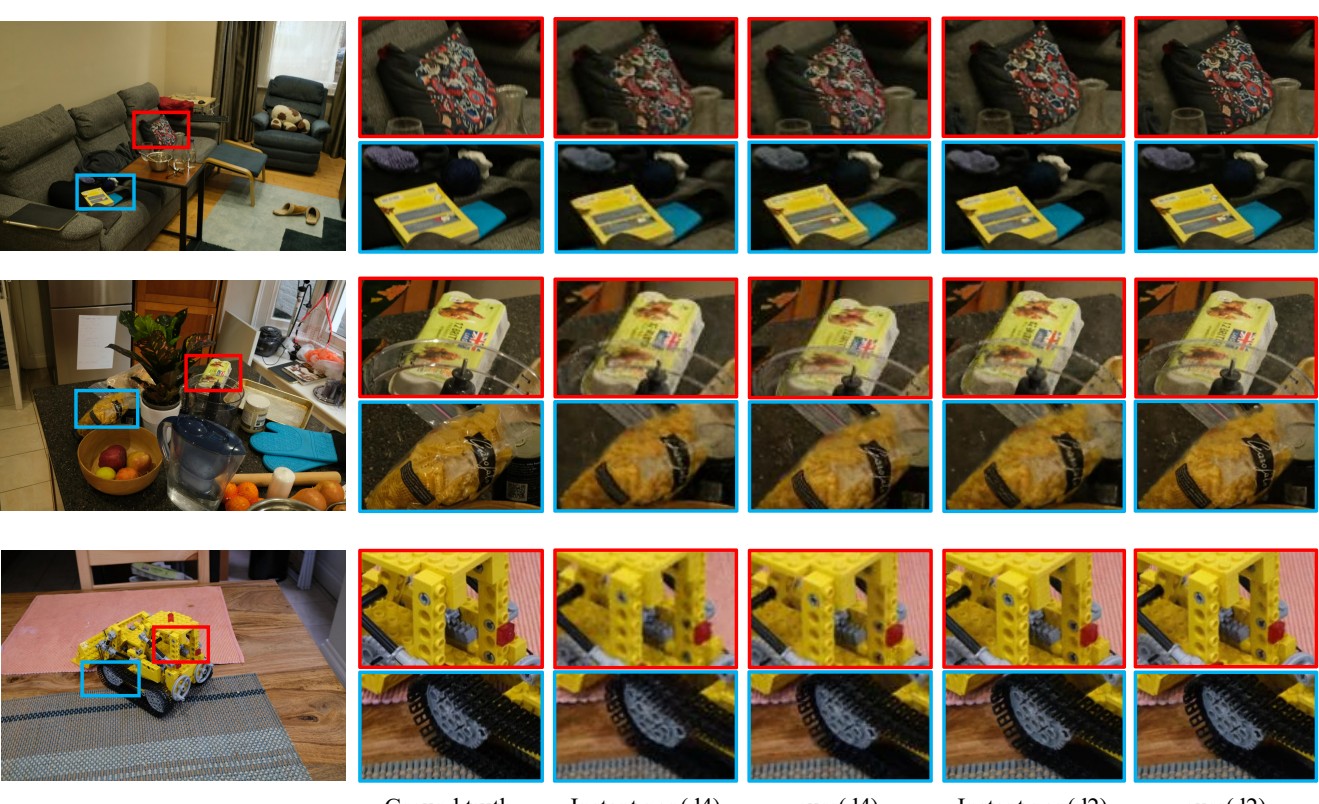

Ground truth          Instant ngp(d4)          ours(d4)          Instant ngp(d2)          ours(d2)

**Figure 5: The experimental results graph for MipNeRF. We compared our visualization results with Instant NGP at two downsampling rates: 2x and 4x. Our method consistently produces higher-quality images at both resolutions, with a more pronounced effect at higher resolutions."**

slight increase in training time, posing no significant computational burden.

## 4.5 TanksAndTemple

The Tankandtemple dataset provides eight scenes at a resolution of 1080p. We tested our method on four bounded scenes within the dataset. As shown in Table 3, our method achieves state-of-the-art performance in several scenes provided by the dataset. Figure1 in the paper demonstrates the convergence speed and final rendering results of our method compared to the baseline. We observe faster

convergence and better image quality with our method. Figure 6 illustrates the rendering results of our method at the edges, where we capture high-frequency information more effectively, resulting in clearer rendered images.

## 4.6 Ablation studies

Our method employs a two-stage segmentation approach, aiming to balance the influence of color and error on segmentation. To validate the effectiveness of our method, we separately test the results of segmentation based solely on RGB color and solely on

(b) Mip NeRF 360.

| Method | Mip NeRF down 4 | | | | Mip NeRF down 2 | | | |
| | PSNR↑ | SSIM↑ | LPIPS↓ | Times↓ | PSNR↑ | SSIM↑ | LPIPS↓ | Times↓ |
| --- | --- | --- | --- | --- | --- | --- | --- | --- |
| DVGO | 25.65 | 0.705 | 0.381 | 626 | 25.15 | 0.668 | 0.463 | 654 |
| DVGO(ours) | 26.11 | 0.731 | 0.372 | 641 | 25.62 | **0.671** | **0.442** | 701 |
| instant ngp | 25.27 | 0.702 | 0.376 | 274 | 24.83 | 0.613 | 0.482 | 290 |
| instant ngp(ours) | **27.37** | **0.733** | **0.361** | 281 | **25.81** | 0.634 | 0.462 | 299 |

Table 2: These are the quantitative results for the MipNeRF dataset. As demonstrated, our method shows performance improvements in the majority of the scenes.

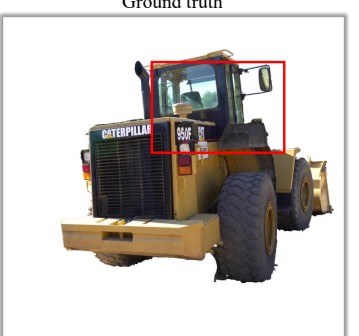
Ground truth

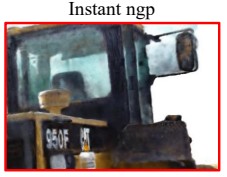
Instant ngp

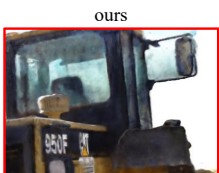
ours

Figure 6: We compare the reconstruction results of our method and the baseline at the edges, where our method provides clearer edge information.

| Method | Barn | Caterpillar | Family | Truck | Mean |
| --- | --- | --- | --- | --- | --- |
| dvgo | 26.87 | 25.70 | 33.73 | 27.08 | 28.35 |
| dvgo(ours) | **27.01** | **25.92** | **33.77** | **27.69** | **28.60** |
| torch ngp | 26.18 | 25.04 | 31.87 | 26.67 | 27.44 |
| torch ngp(ours) | 26.39 | 25.35 | 32.76 | 26.82 | 27.83 |

Table 3: These are the quantitative results for the Tankand-Tamplate dataset. As demonstrated, our method shows performance improvements in the majority of the scenes.

rendering error. We use Instant NGP as the baseline and test on the NeRF synthetic dataset. As shown in the table 4, segmentation based solely on RGB color leads to unstable results. While it achieves good performance in some scenes, the results in certain scenes become unacceptable. On the other hand, using error-based segmentation alone results in relatively stable improvements, but areas with large errors may overlook structural information in the scene. Therefore, our two-stage sampling strategy strikes a good balance between stability and rendering quality.

## 5 CONCLUSION

In this paper, we introduce a method called SPS which employs superpixels to guide the pixel-level ray sampling process during NeRF training, allowing for the processing of large-scale image inputs

| Method | chair | drums | ficus | hotdog | lego |
| --- | --- | --- | --- | --- | --- |
| instant ngp | 29.06 | 24.86 | 29.87 | 33.98 | 30.40 |
| instant ngp(rgb) | 27.53 | **26.10** | 28.92 | 34.01 | **32.13** |
| instant ngp(error) | 29.61 | 25.53 | 31.87 | 33.96 | 31.12 |
| instant ngp(ours) | **29.65** | 25.57 | **30.30** | **34.24** | 31.07 |

Table 4: The experimental results graph for SLIC segmentation. We conducted experiments testing the results of segmentation based on RGB, rendering error, and our two-stage segmentation method. It can be observed that our method achieves the best performance in terms of both stability and effectiveness.

and the generation of high-quality rendered images. Leveraging the characteristics of superpixels, we devise a novel segmentation, updating, and sampling strategy tailored to them. Our strategy effectively balances the relationship between sampling errors and color information, enabling more efficient information extraction from large-scale inputs. Additionally, we propose a new loss function that better utilizes the characteristics of superpixels. In experimental evaluations, our approach achieves state-of-the-art performance compared to existing methods.

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
