# OpenReview forum: "Superpixel-based Efficient Sampling for Learning Neural Fields from Large Input"
_acmmm.org/ACMMM/2024/Conference — MM2024 Poster_

### Official Review · Reviewer_82Xx · 2024-05-21

**Rating:** 3
**Confidence:** 2

**Summary:**

A super-pixel efficient sampling method (SES) is introduced to sample large input data.

**Strengths:**

The SES method optimizes pixel-level ray sampling by segmenting the error map into multiple superpixels using the slic algorithm, and dynamically updating their errors during training to increase ray sampling in areas with higher rendering errors.
The authors conduct some experiments to evaluate the effectiveness of their method.

**Limitations:**

However, there are still some shortcomings in the paper.

1. In the case that there is still space left in the paper, the visual comparative experiment is still insufficient, and there is no supplementary material to show more experimental effects.
2. There is no ablation experiment of the proposed loss function and segmenting the image into superpixels.

**Suitability:**

2

---

### Official Review · Reviewer_P7CC · 2024-05-24

**Rating:** 4
**Confidence:** 3

**Summary:**

This work proposes an efficient and effective sampling method for learning neural fields from large inputs. It utilizes superpixels to guide the pixel-level ray sampling process during NeRF training.

**Strengths:**

The visualization results seem impressive, and the motivation for employing a superpixel-based approach is reasonable.

The writing is clear and well-articulated.

**Limitations:**

1. Could you provide comparisons of efficiency changes between your method and quadtree sampling as the resolution increases? This would further highlight the effectiveness of your proposed method.

2. In Table 4, why does instant ngp(rgb) outperform your proposed method by approximately 1dB?

3. There are many hyperparameters involved; are there any ablation studies concerning them?

4. There are typographical errors. For example, "33 neighborhood" (Line 324-325) and "SLIC[?]" algorithm (Line 372-373).

5. I suggest including more citations and discussions of relevant MM conference papers on 3D to enhance the article's relevance for this conference. For example, "Geometry-Aware Reference Synthesis for Multi-View Image Super-Resolution" and "Space-Angle Super-Resolution for Multi-View Images."

**Suitability:**

2

---

### Official Review · Reviewer_DK7d · 2024-06-08

**Rating:** 4
**Confidence:** 4

**Summary:**

This paper introduces a novel method called Superpixel-based Efficient Sampling(SES) to enhance the efficiency and accuracy of learning neural fields, particularly for novel view synthesis tasks. SES effectively addresses challenges such as redundant sampling of rays and slower convergence on high-resolution images by optimizing light ray sampling. It leverages superpixels to reduce redundancy and concentrate on areas with significant rendering errors.

**Strengths:**

The use of superpixel segmentation via SLIC algorithm to partition the image into distinct local information units is a novel approach, supported by a robust theoretical framework including color space conversion. The SES method demonstrates superior performance in reconstructing high-frequency details as the resolution of synthesized view increases. This evidenced both quantitatively and qualitatively through experiments.

**Limitations:**

Despite the focus on efficient and effective sampling for learning neural fields, the paper lacks sufficient quantitative evaluation regarding efficiency metrics, such as training time. Additionally, it does not analyze the increase in training time when applied to the MipNeRF dataset.

**Suitability:**

2

---

### Meta-Review · Area_Chair_GA2x · 2024-07-01

**Recommendation:** Accept (Poster)
**Confidence:** 4

**Metareview:**

Authors have proposed novel method for enhancing the efficiency and accuracy of learning neural fields. The use of superpixel segmentation via SLIC algorithm to partition the image into distinct local information units is a novel approach, supported by a robust theoretical framework including color space conversion. The SES method demonstrates superior performance in reconstructing high-frequency details as the resolution of synthesized view increases. This evidenced both quantitatively and qualitatively through experiments.

The shortcoming of this paper is in its ablation studies. There are lack of result shown in ablation study to show effectiveness of its method compared to other possibilities. Because the limitation of this paper is merely in its ablation study, I am lean toward accepting this paper for a poster session